# Graves’ Disease after mRNA COVID-19 Vaccination, with the Presence of Autoimmune Antibodies Even One Year Later

**DOI:** 10.3390/vaccines11050934

**Published:** 2023-05-03

**Authors:** Fuminori Nakamura, Toru Awaya, Masahiro Ohira, Yoshinari Enomoto, Masao Moroi, Masato Nakamura

**Affiliations:** 1Department of Cardiovascular Medicine, Toho University Ohashi Medical Center, 2-22-36 Ohashi, Meguro-ku, Tokyo 153-8515, Japan; fuminori.nakamura2@gmail.com (F.N.);; 2Department of Diabetes, Metabolism and Endocrinology, Toho University Ohashi Medical Center, Tokyo 153-8515, Japan

**Keywords:** ASIA, cross-reactivity, Graves’ disease, mRNA COVID-19 vaccination

## Abstract

A 45-year-old man who had received his second mRNA COVID-19 vaccination one week earlier was presented to the emergency department with chest discomfort. Therefore, we suspected post-vaccination myocarditis; however, the patient showed no signs of myocarditis. After 2 weeks, he revisited the hospital complaining of palpitations, hand tremors, and weight loss. The patient exhibited high free thyroxine (FT4) (6.42 ng/dL), low thyroid-stimulating hormone (TSH) (<0.01 μIU/mL), and high TSH receptor antibody (17.5 IU/L) levels, and was diagnosed with Graves’ disease. Thiamazole was administered, and the patient’s FT4 levels normalized after 30 days. One year later, the patient’s FT4 is stable; however, their TSH receptor antibodies have not become negative and thiamazole has continued. This is the first case report to follow the course of Graves’ disease one year after mRNA COVID-19 vaccination.

## 1. Introduction

Since the onset of the COVID-19 pandemic, many people have been infected and have died around the world. As of April 2023, 763,740,140 confirmed cases and 6,908,554 deaths worldwide have been reported to the World Health Organization [1].

Under such circumstances, although the COVID-19 mRNA vaccine was developed and administered worldwide, various adverse effects have been reported, including cardiovascular diseases such as acute coronary syndrome (ACS) and myocarditis [2,3,4], and hyperthyroidism including Graves’ disease, Graves’ orbitopathy, subacute thyroiditis, and silent thyroiditis [5,6,7,8,9,10,11,12,13,14].

Expected cases of Graves’ disease in 2021 increased markedly compared to 2017–2019. Two-thirds of patients had a history of vaccination in the 90 days prior to symptom onset [11]. In healthcare workers, a gradual increase in the mean thyroid-stimulating hormone (TSH) receptor antibody (TRAb) in response to frequent vaccination has been reported [14].

Graves’ disease is a disorder caused by autoantibodies against the TSH receptor in the thyroid. Unlike most autoantibodies, which are inhibitory, this one is stimulatory, resulting in excessive synthesis and the secretion of T4 and T3.

The main cause of Graves’ disease following COVID-19 vaccination is related to autoimmunity, which is explained by autoimmune/inflammatory syndrome induced by adjuvants (ASIA) and cross-reactivity between thyroid tissue including thyroid peroxidase (TPO) and the SARS-COV2 spike protein [5,6,7,8,9,10,11,12,13,14,15]. The mRNA vaccine contains adjuvants such as lipid nanoparticles (LNPs) that can trigger ASIA [16,17].

Reports of post-treatment outcomes for vaccine-related Graves’ disease are limited. Here, we report a case of new-onset Graves’ disease after receiving the COVID-19 vaccine.

## 2. Case Presentation

A 45-year-old man visited the outpatient clinic with chest discomfort, and he was referred to the hospital with suspected post-vaccination myocarditis. He had no allergies and no family history or history of autoimmune disease, including thyroid disease. The patient received his second mRNA COVID-19 (Moderna) vaccination one week before symptom onset. A sinus tachycardia (116 bpm) without ST–T segment changes was noted on the electrocardiogram (ECG) (Figure 1B). Blood examination and a transthoracic echocardiogram showed no signs of myocarditis. After two weeks, he revisited the hospital complaining of palpitations, hand tremors, and weight loss. Graves’ disease was diagnosed based on measurements of high free triiodothyronine (FT3) (27.5 pg/mL), high free thyroxine (FT4) (6.42 ng/dL), low TSH (<0.01 μIU/mL), and high TRAb (17.5 IU/L) levels. A thyroid echography demonstrated diffuse swelling of the thyroid gland and an uneven internal hypoechoic image suggestive of a diffuse goiter consistent with Graves’ disease (Figure 1A) [18,19]. Two years earlier, blood tests showed normal thyroid function (TSH 1.07 IU/L, FT4 1.31 ng/dL) (Figure 1B). One month after thiamazole (30 mg/day) and bisoprolol (2.5 mg/day) were administered, the FT3 and FT4 levels improved to 4.3 pg/mL and 0.95 ng/dL, respectively. The patient’s symptoms and heart rate also became normalized (Figure 1B). Four months after treatment, TRAb transiently increased from 17.5 to 30.9 IU/L despite adequate thyroid function control (Figure 1B). After one year, the dosages of thiamazole and bisoprolol were reduced from 30 mg to 7.5 mg and from 2.5 mg to 0.625 mg, respectively. TSH of 3.5 μIU/mL, FT3 of 3.2 pg/mL, and FT4 of 1.1 ng/dL levels stabilized, and TRAb decreased from 17.5 to 6.3 IU/L (the peak level was 30.9 IU/L after four months), however, it did not become negative (Figure 1B).

## 3. Discussion

In this case, we report a case of suspected Graves’ disease due to COVID-19 vaccination with a good response to thiamazole, but TRAb was still positive one year later. Although Graves’ disease following COVID-19 vaccination has been reported [5,6,7,8,9,10,11,12,13,14], few reports have presented data on the chronic phase of Graves’ disease after vaccination.

There are two types of TRAb including TSH-stimulation blocking antibody (TSBAb) and thyroid stimulating antibody (TSAb), and TRAb is positive even when the former is positive [20], and TSBAb causes hypothyroidism and thyroid atrophy [21]. In this case, TSBAb was considered to be negative because of hyperthyroidism and no thyroid atrophy at echography (Figure 1A).

The present case is a 45-year-old male who was diagnosed with Graves’ disease after his second vaccination. A recent study indicates that among patients with new-onset Graves’ disease, those with onset within four weeks of COVID-19 vaccination were older (median age 51 years vs. 35 years) and more likely to be male (40.0% vs. 13.6%). Moreover, the incidence of Graves’ disease after vaccination was 25% for the first dose, 65% for the second, and 10% for the third, based on data from January to December 2021 [12]. The age, sex, and number of vaccinations in this case were consistent with the results of the study, and so it was a case with typical characteristics. Furthermore, the number of expected cases of Graves’ disease in 2021 has increased markedly compared to 2017–2019. A total of 44/66 (66.7%) had a history of vaccination in the 90 days prior to symptom onset [11].

With regard to treatment for Graves’ disease after vaccination, the initial therapeutic response was reported as good, with 40% of patients being TRAb negative (TRAb ≤ 1.75 IU/L) after 3 months. It has been speculated that the vaccine-induced autoimmune process in these patients may be transient or self-healing [12]. However, TRAb transiently increased from 17.5 to 30.9 IU/L in the present case and remained elevated one year later despite adequate thyroid function control. Therefore, this differs from the previous report (Figure 1B).

A high TRAb (>12 IU/L) at diagnosis and/or a positive TRAb (>1.5 IU/L) at discontinuation of treatment has been associated with a high likelihood of relapse. In particular, it has been reported that many recurrences occur within 2 years, and therefore special attention should be paid during this period [22]. Huang et al. also reported a decrease in TRAb in the appropriately treated group compared to the inappropriately treated group [23]. Thus, although the initial therapeutic effect was very favorable in this case, interruption of treatment needs to be completed with caution. In addition, the trend of TRAb measurements may be informative in the course of Graves’ disease after vaccination.

The main cause of Graves’ disease following COVID-19 vaccination is related to autoimmunity including TRAb [5,6,7,8,9,10,11,12,13,14]. ASIA has previously been reported to cause autoimmune diseases after papillomavirus, influenza, and hepatitis B virus vaccination (Figure 2A). While adjuvants are vaccine components to activate the immune system, they may cause ASIA. Aluminum salt is a common adjuvant frequently used in vaccines, such as for papillomavirus, hepatitis B virus, and pneumococcal conjugate vaccines [5,6,13,14,15]. The mRNA vaccination contains adjuvants such as lipid nanoparticles (LNPs), which produce inflammatory cytokines including interleukin (IL)-1β and IL-6, and can trigger ASIA [16,17]. LNPs release excess IL-1β(22×) and IL-6 (≥12) while producing more antibodies than AddaVax (an MF59-like adjuvant) [16]. Additionally, thyroid tissues have angiotensin-converting enzyme 2 (ACE2) expression [5]. Spike protein produced by the COVID-19 mRNA vaccine binds to ACE2 and induces ACE2 downregulation, which causes a release of IL-1β and IL-6 [24]. The spike protein itself also induces IL-1β and IL-6 [25]. IL-1β and IL-6 may induce autoimmune disease due to the activation of T helper 17 cells and suppression of regulatory T cells [10,26]. There have also been reports on genetic susceptibility to ASIA syndrome, including carriers of the HLA DRB1 haplotype [15].

Another cause of autoimmunity is explained by the cross-reactivity theory (Figure 2B). There is amino acid sequence homology (molecular mimicry) between TPO and the SARS-COV2 spike protein [9,10]. Therefore, cross-reactivity between the thyroid tissue and spike protein produced by the mRNA vaccine may produce antibodies against the thyroid antigen. Additionally, a gradual increase in the mean TRAb in response to frequent mRNA vaccination in Japanese healthcare workers has been reported [14]. This may be the result of cross-reactivity with thyroid tissue caused by the high production of spike protein due to frequent vaccination. Although TPO was not measured in the present case, it may be more effective to measure TPO in addition to TRAb in post-vaccine Graves’ disease.

Moreover, a relationship between age-associated B cells (ABCs) and autoimmunity after mRNA vaccination has recently been suggested [27]. ABCs expand continuously with age and induce IL-4/IL-10 secretion and Th17 induction, which are associated with autoimmune diseases. Toll-like receptor (TLR)-7 and TLR-9 increase ABCs activity [28]. TLR-7/8 and TLR-9 are stimulated by mRNA and DNA vaccines [27]. Hence, vaccine-activated ABCs may be involved in autoimmunity [6,27,28]. Reportedly, frequent mRNA vaccination increases IgG4, and IL-4/IL-10 are involved in the class switch to IgG4 [29]. IgG4 is involved in autoimmunity, inflammation, and fibrosis [30]. IgG4-related autoimmune pancreatitis [31], IgG4-related lung disease [32], and relapse of IgG4-related nephritis [33] have been reported after mRNA vaccination. In addition, unlike IgG1 and IgG3, IgG4 has a low effector function and little antiviral activity [29].

Vaccine-related autoimmune diseases other than Graves’ disease have been reported, including Guillain–Barré syndrome, neuromyelitis optica, vaccine-induced thrombotic thrombocytopenia, autoimmune liver disease, immune thrombocytopenic purpura, IgA nephropathy, IgG4 nephritis, systemic lupus erythematosus, autoimmune polyarthritis, rheumatoid arthritis, type 1 diabetes mellitus, and vasculitis [4,6,10].

Risk factors for Grave’s disease include genetic predisposition and interactions between endogenous (estrogens, X-inactivation, and microchimerism) and environmental (smoking, iodine excess, selenium, vitamin D deficiency, and occupational exposure to Agent Orange) factors [34]. Genetic predisposition of human leukocyte antigen (HLA) B* 35, HLA C* 04, and HLA-A* 11 has been implicated in SARS-CoV-2 vaccine-induced subacute thyroiditis [13]. Even in patients with a history of Graves’ disease, recurrence of Graves’ disease following COVID-19 vaccination has been reported. Special caution is required in patients with a history of Graves’ disease [7,13]. The patient had no family history or history of autoimmune disease, including thyroid disease and no other endogenous or environmental factors or infections. Symptoms of palpitations, weight loss, and hand tremors were presented after the second vaccination. Considering the lack of risk factors and the appearance of the symptoms after vaccination, the association between worsening hyperthyroidism and vaccination was highly suggested.

Graves’ disease is a rare adverse reaction compared with ACS and myocarditis [2,7]; therefore, it may be underdiagnosed. This case was also referred to our hospital on suspicion of myocarditis because of chest discomfort after the COVID-19 vaccination. In addition, thyroid crisis can also cause takotsubo cardiomyopathy and fatal arrhythmias, which may further complicate differentiation from vaccine-related cardiovascular disease. The onset post-vaccination symptoms of Graves’ disease was 2–20 days [5]. In contrast, post-vaccination ACS and myocarditis cases developed symptoms, and the medians were 1 and 3 days, respectively [3]. Differential diagnoses of post-vaccination chest discomfort have been reported [4]. Hyperthyroidism should be considered when a patient presents sinus tachycardia, no ST change on an ECG, and normal troponin levels after mRNA COVID-19 vaccination. Older patients with hyperthyroidism following vaccination can be presented with atrial fibrillation, heart failure, and Graves’ orbitopathy, which require special care [13]. Even if the response to the treatment of post-vaccine Graves’ disease is good, it is important to monitor the progress of TRAb.

## 4. Conclusions

We present a male patient with Graves’ disease with the presence of TRAb even one year after mRNA vaccination. In the case of chest discomfort following the COVID-19 vaccination, we recommend assessing FT4 and TSH in addition to troponin in consideration of the possibility of thyroid disease. In general, Graves’ disease is more common in women and those with a history of autoimmune disease. However, post-vaccine Graves’ disease has been reported in males without a history of autoimmune disease (such as in the present case) and requires caution. Long-term data on post-vaccine Graves’ disease are unknown and require careful follow-up.

## Figures and Tables

**Figure 1 vaccines-11-00934-f001:**
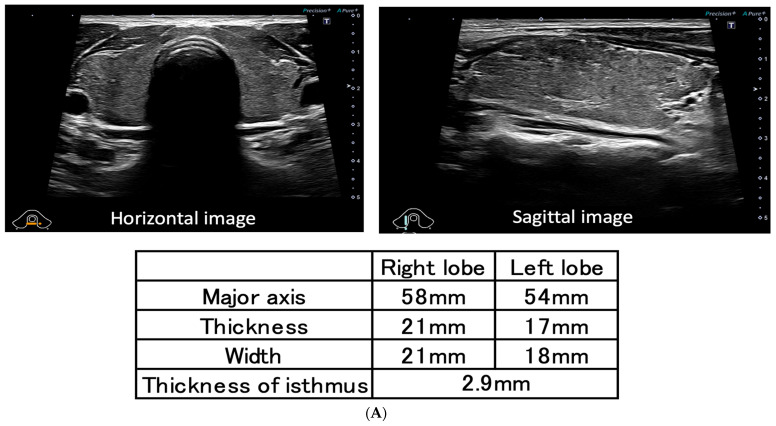
(**A**) Thyroid echography demonstrates diffused swelling of the thyroid gland and an uneven internal hypoechoic image suggestive of diffuse goiter; (**B**) TRAb was at high levels at diagnosis (17.5 IU/L), and the peak level was 30.9 IU/L after four months. HR, TSH, FT3, FT4, and TRAb levels have improved since thiamazole was administered; however, TRAb has not yet become negative (6.3 IU/L) after one year. HR: heart rate; MMI: thiamazole; TSH: thyroid-stimulating hormone; FT3: free triiodothyronine; FT4: free thyroxine; TRAb: TSH receptor antibody.

**Figure 2 vaccines-11-00934-f002:**
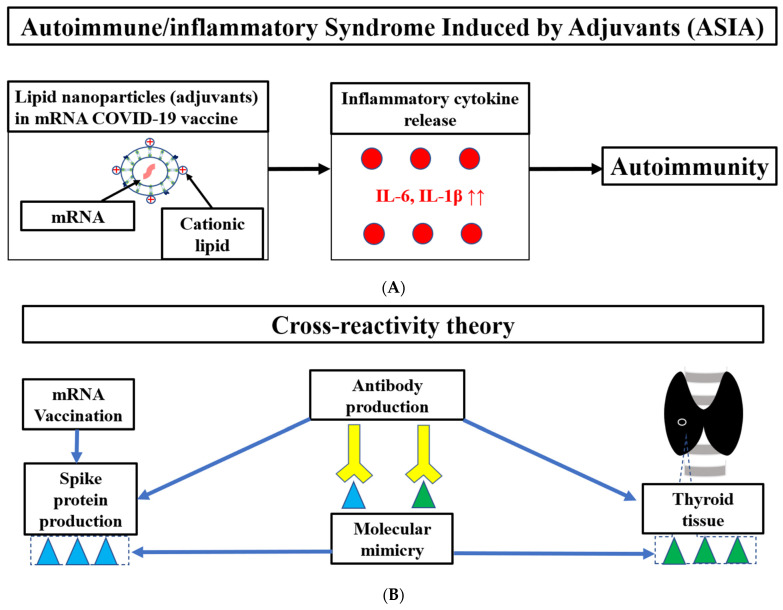
The potential mechanisms including ASIA (**A**) and the cross-reactivity theory (**B**) of Graves’ disease following mRNA vaccination in this case. mRNA, messenger RNA; IL, interleukin.

## Data Availability

No new data were created or analyzed in this study. Data sharing is not applicable to this article.

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
