# Peer review of "Graves’ Disease after mRNA COVID-19 Vaccination, with the Presence of Autoimmune Antibodies Even One Year Later"

_vaccines, 2023, doi:10.3390/vaccines11050934_

Round 1
Reviewer 1 Report
Comments and Suggestions for Authors:
Dear Editor-in-Chief and authors,
I read the manuscript “Graves' disease after mRNA COVID-19 vaccination, with the presence of autoimmune antibodies even one year later” with great interest. The study is interesting and the manuscript is well written.
The abstract is well structured showing the side effect of COVID-19 Vaccination. Further comments and suggestions are follows:
Revision: Major
Ø No previous history of the patient was mentioned related to TRAb positivity or hyperthyroidism.
Ø Discuss other factors responsible for TRAb positivity which is the indicator of Graves’ disease.
Ø Why TPOAb test was not done which is required for thyroid autoimmunity.
http://dx.doi.org/10.1136/jcp-2022-208290
Ø Family history was not recorded which is also a risk factor. The authors are unable to mention it.
Ø There are other environmental factors responsible for Graves’ disease e.g Iodine excess, Selenium, Vit. D and Agent Orange.
https://doi.org/10.1016/j.beem.2020.101387
Ø It is suggested for the authors to describe reasons of positivity of TRAb and other factors in the discussion.
Reviewer 2 Report
Dear Editor,
The authors present the case of a patient developing Graves‘s disease following COVID vaccination with mRNA vaccine. Similar reports have been published and therefore this manuscript is not unique. One important feature of the case seems to be the persistence of autoantibiodies after one year, nevertheless the authors fail to a certain level to communicate whether this is particular to this case and how this adds to the current literature. This should be reinforced with comparisons with the literature. In addition the introduction should add information regarding the normal development of the disease.
Pharmacovigilance and the reporting of unexpected side effects of vaccines is of extreme importance due to the preventive character of vaccination. Respective to that matter, I believe the authors do a nice work justifying why these case could be a consequence of vaccination..
In addition, some minor points that could be improved
- the authors describe an improvement in TSH after beginning of therapy, nervertheless it persists at <0,01... i can see that the other parameters have improved, however it can be confusing for the readers
- i don't think that discribing atrial premature contractions two years previous to the case adds any relevance to the manuscript
- the ECG image is not necessary, you can summarize it adding the hear rate to the image in C. It is enough to describe it as a sinus rhythm.
- in the conclusion the authors should realy focus in the particular features of this case and how this adds value to this particular topic in general. Recommendations are normally made with panels of experts and follow spefic guidelines. Maybe it could be rephrased to better suit the purpose of this manuscript.
Reviewer 3 Report
In the following case report entitled "Graves' disease after mRNA COVID-19 vaccination, with the presence of autoimmune antibodies even one year later". The idea of the study design is good.
It's better if the introduction section will be elaborated. Discuss the mechanism of Graves' disease a bit more, and if can add a diagram it would be great. Results and discussion are well explained and presented.
Minor spelling and English Grammar mistakes are there, please read it carefully and improve it before resubmission.
Reviewer 4 Report
-
The case presentation lacks important information that should be mentioned, such as whether the patient had any pre-existing conditions or risk factors for autoimmune diseases, which could potentially affect the validity of the conclusions. Furthermore, the authors should provide more context for the patient's medical history, including any relevant family history.
-
The figures should be clearly labeled and numbered, and the captions should provide more detail on the methods and results shown. The legends for Figure 1A, 1B, and 1C should be more explicit about which panels correspond to each result. Additionally, the authors should consider including a timeline to help visualize the events surrounding the patient's diagnosis and treatment.
-
The discussion section should provide a more comprehensive review of the existing literature on vaccine-induced Graves' disease and address any inconsistencies or gaps in knowledge. This would help contextualize the case report within the broader scientific understanding of the phenomenon.
-
The authors should clarify the basis for their conclusion that the mRNA COVID-19 vaccination was the cause of Graves' disease in this case. While the presentation of symptoms following vaccination suggests a correlation, it does not necessarily imply causation. Alternative explanations should be considered and discussed.
-
In the conclusion section, the authors should provide more specific recommendations for healthcare providers regarding the assessment of potential vaccine-induced Graves' disease. For instance, they should discuss which patients might be at higher risk and when further investigation might be warranted.
-
The paper needs to be thoroughly proofread for grammatical errors, inconsistencies in abbreviations, and formatting issues.
Round 2
Reviewer 1 Report
Dear Editor, The comments have addressed and it is accepted now.
Reviewer 4 Report
I extend my gratitude to the authors for their prompt response. I confirm that I am satisfied with the publications related to this manuscript.